# A Fast Design Method of Anisotropic Dielectric Lens for Vortex Electromagnetic Wave Based on Deep Learning

**DOI:** 10.3390/ma16062254

**Published:** 2023-03-10

**Authors:** Bingyang Liang, Yonghua Zhang, Yuanguo Zhou, Weiqiang Liu, Tao Ni, Anyi Wang, Yanan Fan

**Affiliations:** 1College of Communication and Information Engineering, Xi’an University of Science and Technology, Xi’an 710054, China; 2National Key Laboratory on Vacuum Electronics, University of Electronic Science and Technology of China (UESTC), Chengdu 610054, China; 3The Xi’an Research Institute of Navigation Technology, Xi’an 710054, China; 4The National Space Science Center, Chinese Academy of Sciences, Beijing 100190, China

**Keywords:** orbital angular momentum, vortex electromagnetic wave, anisotropic dielectric lens, deep learning neural network (DNN)

## Abstract

Orbital angular momentum (OAM) has made it possible to regulate classical waves in novel ways, which is more energy- or information-efficient than conventional plane wave technology. This work aims to realize the transition of antenna radiation mode through the rapid design of an anisotropic dielectric lens. The deep learning neural network (DNN) is used to train the electromagnetic properties of dielectric cell structures. Nine variable parameters for changing the dielectric unit structure are present in the input layer of the DNN network. The trained network can predict the transmission phase of the unit cell structure with greater than 98% accuracy within a specific range. Then, to build the corresponding relationship between the phase and the parameters, the gray wolf optimization algorithm is applied. In less than 0.3 s, the trained network can predict the transmission coefficients of the 31 × 31 unit structure in the arrays with great accuracy. Finally, we provide two examples of neural network-based rapid anisotropic dielectric lens design. Dielectric lenses produce the OAM modes +1, −1, and −1, +2 under TE and TM wave irradiation, respectively. This approach resolves the difficult phase matching and time-consuming design issues associated with producing a dielectric lens.

## 1. Introduction

In recent years, researchers have been paying close attention to the vortex beam carrying orbital angular momentum (OAM) because it offers a fresh approach to boosting the wireless communications channel count and improving imaging resolution [1,2]. The orbital angular momentum of the vortex electromagnetic wave is realized primarily through the spiral phase characteristic in the orthogonal direction with the electromagnetic wave propagation axis [3,4]. Due to the spiral phase distribution and the mutually orthogonal OAM mode, vortex electromagnetic waves show great potential in many fields, such as particle manipulation [5,6], optical and electromagnetic imaging [7,8,9], communication [10,11,12,13], etc.

The vortex electromagnetic wave technology offers a fresh approach to the problems of channel congestion and insufficient spectrum resource use. Spiral reflectors [14,15], antenna arrays [1,16], spiral phase plates [17,18], and metasurfaces [19,20] can all produce vortex waves. However, only a single OAM mode of vortex wave may be realized for antennas with a certain configuration; therefore, vortex wave multiplexing cannot be realized well. Array antennas can produce a variety of OAM modes by using electronically controlled phase shifters and attenuators, but this requires sophisticated feed systems and pricey active components. Multiple OAM modes can be produced by both phased array and reconfigurable antennas at the same aperture, but may need be switched at separate times [21]. This time-sharing strategy reduces the efficiency of vortex wave transmission and detection.

Although various application circumstances call for various design topologies, there is one general rule in the design of generating OAM: the azimuthal phase term is inserted into the electromagnetic wave. Aperture super surface [22], dielectric surface [23], dielectric lens [24,25], and other techniques can be used to adjust the phase change for antennas with certain shapes to produce various vortex wave patterns. However, the design process depends on software simulation and numerical simulation, and it also depends on ongoing optimization or empirical reasoning. As a result, numerous full-wave numerical computations are necessary, and although the results of the calculations are accurate, the design efficiency is poor, and extensive computational resources are needed. Typically, only one mode of the vortex wave can be realized once the lens structure has been established. An anisotropic lens construction can be utilized to realize more vortex wave modes to enhance the vortex wave’s performance [26,27]. However, during the anisotropic structure design process, more dielectric unit structures need to be estimated [28,29]. The typical full-wave electromagnetic computation method is time-consuming and resource-consuming.

Deep learning has demonstrated exceptional benefits in applications such as computer vision [30,31], natural language processing [32,33], speech recognition [34,35], and face identification [36,37]. The huge potential of application in algorithm optimization, in particular, has brought us new methodologies. Maokun Li’s group proposed an encoding approach based on a fully convolutional neural network to produce programmable metasurface complex beams. In less than one millisecond, the network can compute the encoding matrix that matches the input requirements [38]. Sensong An and colleagues demonstrated a neural network modeling technique for describing three-dimensional dielectric structures that significantly increased the speed and accuracy of defining subwavelength optical phenomena [39,40]. Qian Zhang et al. proposed using machine learning to design one-bit anisotropic digital coding metasurfaces. It is very convenient to realize the left and right circularly polarized medium metasurfaces [41]. Roberts et al. proposed a deep learning technique for the forward and inverse plasmonic metasurface structure designs. et al. proposed a deep learning technique for the forward and inverse plasmonic metasurface structure designs [42]. Li Jiang et al. demonstrated the neural network’s ability to process six geometric parameters to precisely anticipate the phase, and they achieved accurate forward prediction and inverse design of nanoprotein metasurfaces [43]. More research has revealed that neural networks can quickly design phase surfaces and have considerable application potential in forecasting the phase of metasurfaces and dielectric surfaces [44]. The machine learning algorithm used above to build metasurfaces has produced successful outcomes, and the needed cell can be designed in milliseconds. However, there has not been any work to bring the machine learning-based design process into practice for anisotropic vortex electromagnetic dielectric lenses that need more dielectric unit structures.

In this work, we built a deep learning neural network that can realize the rapid design of multiple anisotropic dielectric lenses to realize additional modes of vortex wave in the same dielectric structure and improve its communication capacity. To quickly design the demand phase, we employ DNN networks to learn and optimize the unit structure with nine variables. The gray wolf optimization approach is utilized to achieve quick dielectric lens unit positioning. Finally, two vortex wave multiplexed dielectric lens are provided as a last step.

## 2. Deep Learning Neural Network Establishment and Training

We designed the dielectric unit structure depicted in Figure 1a to achieve the multiplexing of waves in various modes. The unit structure consists of the rectangular dielectric cylinder on both sides of a dielectric substrate, where the width of the dielectric substrate w= 7.5 mm, height $h_{3}=$ 2.0 mm. The width of the three dielectric columns on the dielectric substrate’s front side are x1, x2, x3, and the height is h1, the thickness is w/3 = 2.5 mm. The breadth of the three dielectric columns on the substrate’s rear side is x3, x4, and x5, correspondingly, and the height is $h_{2}$. The three dielectric columns on the media substrate’s reverse side have corresponding widths of x3, x4, and x5, the height is h2. Polylactic acid (PLA), which has a relative dielectric constant of 2.67, is used as the dielectric medium to enable 3D printing. The transmission coefficient of the unit structure at 10 GHz–15 GHz is shown in Figure 1b. This unit structure’s TE wave and TM wave transmissivity can reach 90% in the 10 GHz–14 GHz range. The phase change of the dielectric column can range from 0 to 360 when its height is 10 mm to 25 mm. The transmission amplitude and phase of the TE wave and the TM wave differ because of the differences between the dielectric cylinders on each side of the dielectric substrate.

There are eight variables for the basic dielectric unit mentioned above. To attain a specific performance, it is required to optimize the eight factors based on experience during the design process. Therefore, to understand and forecast the spatial phase distribution of dielectric elements, we designed a DNN network model based on the phase component. We boost the response of the TE wave and TM wave to represent its anisotropy. The specific DNN network architecture is shown in Figure 2. We supply the eight-dimensional unit-cell parameters and mode response (TE or TM) as input to the network so that input=[x1,x2,x3,x4,x5,x6,h1,h2,mode]. The distribution probability of each phase response of the unit structure under TE or TM waves is represented by the network output. The cross-entropy loss function maps each dimension to the likelihood of each phase [45,46]. The highest output probability is the phase response.

The activation functions of the first and last layers of the hidden layer, namely hidden layer 1 and hidden layer 9, are finally adjusted to hyperbolic tangent (tanh) function by numerous comparisons and structural modifications [47]. The tanh function is given by
(1)$$tanh=\frac{e^{x}-e^{-x}}{e^{-x}+e^{-x}}.$$

The remaining hidden layers use the leaky ReLU activation function [48]. The activation function and the number of neurons used by each hidden layer in the network are shown in Table 1.

During training, 20,000 groups of unit transmission coefficient data are randomly chosen during the network training procedure to train the network. The network is optimized by using the Adam optimizer. L2 regularizes the parameter weight decay set to $10^{-4}$ to avoid overfitting. The starting learning rate is 0.001. The learning rate drops by 10 times after 20,000 iterations. The maximum allowed number of iterations is 30,000. The results demonstrate that on 20,000 sets of test data, the average phase difference of the DNN model provided in this study is 2.62, and the relative error is less than 1%. On a PC configured with AMD Ryzen5 3600 3.90 GHz, GPU NVIDIA GeForce GTX 1070, sample generation requires 601,200 s, and DNN training takes 291 s.

Examples of the network’s predictions for a few input geometries from the validation set are shown in Figure 3, i.e., the three groups of data in the figure’s geometrical parameters are given by input1 = [5, 3, 3, 4, 3, 5, h1, 14] mm, input2 = [4, 3, 3, 4, 5, 2, h1, 14] mm, input3 = [2, 2, 2, 4, 5, 2, h1, 14] mm. Define the relative error of data (MSE) as follows,
(2)$$MSE=\frac{\left|\right|P_{\mathrm{cal}}-P_{\mathrm{pre}}\left|\right|}{\left|\right|P_{\mathrm{cal}}\left|\right|},$$
where, $P_{\mathrm{cal}}$ and $P_{\mathrm{pre}}$ represent the pahse results of algorithm-based calculation and prediction results, respectively; ||.|| denotes the L2-norm. The MSE of the results in the Figure 3 are 0.0099, 0.0138, and 0.0118. The machine learning network’s prediction accuracy for the medium unit can approach 98%. The accuracy of the neural network we presented can be improved by 2–7% compared to the existing DNN network, and there are more free parameters [41,42,43].

## 3. Reverse Design

In this section, we will examine how machine learning can aid in the reverse design and optimization of metasurfaces structures. There are several ways to use the trained discriminant network to optimize the electromagnetic system, including using the iterative approach to do so directly, incorporating the trained discriminant network into the iterative scheme to do so, etc. The trained discriminant network can simulate the electromagnetic response of the system in a faster order of magnitude than the conventional electromagnetic solver.

In this study, we construct a dielectric lens and realize vortex waves of various modes by using the gray wolf optimization technique and discriminant network. The prey that the gray wolf optimization algorithm needs to seek is the compensating phase that the unit structure at each position in the dielectric lens design requires. The position of each wolf in the wolf pack corresponds to the structural parameters of each dielectric unit, $Z_{i}=\left[x1,x2,x3,x4,x5,x6,h1,h2,Mode\right]$. The goal of the dielectric lens design procedure is to establish each wolf’s ideal position (the structural parameters of each dielectric unit). The evaluation fitness function is utilized throughout the design phase to manage parameter inaccuracy, as follows,
(3)$$fitness=-\left|\left(\varphi_{need}\left(x\right)-\varphi_{i}\left(x\right)\right)+\left(\varphi_{need}\left(y\right)-\varphi_{i}\left(y\right)\right)\right|,$$
where, $\varphi_{need}\left(x\right)$ and $\varphi_{need}\left(y\right)$ represent the phase demand for the respective TE wave (y polarization) and TM wave (x polarization), respectively. $\varphi_{i}\left(x\right)$ and $\varphi_{i}\left(y\right)$ indicate the transmission phase of the cell structure acquired by the DNN network under the TE polarization wave (y polarization) and the TM polarization wave, respectively (x polarization).

### 3.1. Design Method

In this section, the performance of machine learning networks is examined by using dielectric lenses that produce vortex waves in various modes. By varying the phases and amplitude distributions of the dielectric unit structure on the lens surface, the dielectric lens can change the way electromagnetic waves propagate. The dielectric lens must be anisotropic to generate different vortex wave modes when irradiated with different polarized electromagnetic waves. Anisotropy can be produced by various configurations on either side of the dielectric substrate in Figure 1.

To achieve the multiplexing of vortex waves of various modes, the lens can be built in accordance with the phase requirements of vortex waves throughout the design process. The phase distribution at coordinates (x, y) should combine a focusing lens and a vortex plate to produce a vortex beam. The focusing phase distribution is written as [49,50,51]:(4)$$\phi_{tot}\left(x,y\right)=-\frac{2\pi }{\lambda }\left(\sqrt{F_{0}^{2}+x^{2}+y^{2}}-F_{0}\right)+\varphi_{0}+l\tan^{-1}\left(y/x\right).$$

Here, λ wavelength at operating frequency, $F_{0}$ is the focal length, *x* and *y* are the distance between the x-axis and y-axis, respectively, $\varphi_{0}$ is initial focal length, and *l* is the vortex wave mode.

### 3.2. Model 1

First, we present the dielectric lens that produces the mode l=+1 vortex wave when illuminated to TE waves and the mode l=−1 vortex wave when illuminated to TM waves. To reflect the performance of the network, we build a dielectric lens with a 31 × 31 dielectric unit structure. The focal length is set to $F_{0}=150$ mm, the central frequency to 10 GHz. A dielectric lens with 31 × 31 dielectric unit structure necessitates the rapid identification of 961 dielectric unit structure phases. To reduce the phase error and electromagnetic scattering caused by the height difference, the dielectric lens uses a dielectric element of the same height (h1 = 27 mm, h2 = 28 mm, h3=2 mm). Even while the phase under the cell structure may be quickly acquired with the help of a DNN network, finding the requisite cell structure in a large database involves a significant amount of processing. Therefore, it is critical to provide an automatic mapping between the basic element size parameters and the transmission phase to quickly estimate the structure and size of the basic units required to create an anisotropic hypersurface lens antenna. The gray wolf group optimization algorithm is used to locate the structural units matching the phase criteria in the DNN database and combine them to create the dielectric lens in accordance with the phase requirements at the space grid [46,47]. After eliminating the same phase, 510 individual cell structures are still required. Figure 4 demonstrates the calculation compensation phase distribution and process on the dielectric lens. Obtaining the phase of 510 cell structures in a personal computer with an AMD Ryzen5 3600 3.90 GHz CPU takes 30,600.00 s for the FDTD algorithm and 0.02 s for the DNN network.

The FDTD simulation software is used to validate the overall structure, and the results are displayed in Figure 5. The sampling plane measures 248 mm × 248 mm in dimension, and it is 414 mm away from the lens. Figure 5a,b depict the electrical field intensities for the *x* and *y* polarization components. The null values of +1 mode and −1 mode can be clearly seen in the center area. The phase distribution of Figure 5c,d can clearly distinguish two vortex waves with different orders. The transverse field distribution demonstrates that the dielectric lens may generate vortex waves of different modes when irradiated with different polarized electromagnetic waves (TE wave and TM wave). The different OAM modes are dissected by using the Fourier transform analysis to do a quantitative measurement of the purity of the OAM modes, as shown in Figure 5e,f. The following are the related equations [52,53],
(5)$$A_{l}=\frac{1}{2\pi }\int_{0}^{2\pi }\psi (\varphi )e^{-jl\varphi }d\varphi ,$$
(6)$$\psi \left(\varphi \right)=\sum_{l}A_{l}e^{jl\varphi }$$
where ψ(φ) is a characteristic function of the sampled field around the z-axis where the electric field peaks in the sampling plane. Here, the mode of vortex wave is from −7 to +7, and the energy weight of vortex wave is defined as
(7)$$energy-weight=\frac{A_{l}}{\sum_{m=-7}^{7}A_{m}}.$$

Under different polarizations, the purity of the converted vortex wave mode by using the hypersurface is more than 50% (Figure 5e,f). Figure 5 illustrates the structure of the final design, which can be seen as a superposition of two distinct functional metasurfaces that convert TE and TM polarization into the +1 and −1 modes.

### 3.3. Model 2

The DNN method used in this study may easily create combinations of vortex waves with any modes. To further validate network performance and design speed, Model 2 is employed. When subjected to TE wave and TM wave radiation, this model generates −1 and +2 mode numbers. Employ the same quantity of dielectric units (31 × 31) as in the preceding section. The same parameters are used for frequency, focal length and dielectric unit thickness. The calculation compensation phase and process on the dielectric lens as shown in the Figure 6. To remove the same phase, 707 independent cell structures are required. Obtaining the phase of 707 cell structures in a personal computer with an AMD Ryzen5 3600 3.90 GHz CPU takes 42,773.50 s for the FDTD algorithm and 0.23 s for the DNN network. Figure 6d shows the finished dielectric lens construction.

Figure 7 depicts that the electric field distribution and phase distribution of the dielectric lens’s spatial section conform to the morphological distribution characteristics of vortex waves. When TE wave and TM wave are excited, the dielectric lens transforms them into vortex waves with respective mode numbers of −1 and +2. Figure 7e,f illustrates the results of a Fourier transform analysis of the mode purity. First-order vortex wave purity is greater than 0.7, and second-order vortex wave purity is greater than 0.5. When compared to the conventional approach, the performance of the machine learning-designed dielectric lens has surpassed that of the conventional approach in several areas (purity, anisotropic), and it is designed considerably more quickly [26,28].

## 4. Conclusions

In this work, a deep learning neural network-based quick design method for anisotropic dielectric lenses is proposed to address the issues of challenging phase matching and time-consuming design in the process of creating a dielectric lens. To understand and forecast the phase of dielectric elements, a nine-layer DNN neural network based on phase classification is employed. Within a range, the trained network can predict the transmission phase of the unit cell structure rapidly and accurately (more than 98%). The gray wolf algorithm is utilized during the lens design phase to identify the optimal structural unit cell. Two vortex wave lenses design demonstrate the speed and accuracy with which the needed electromagnetic wave transmission mode can be designed by using machine learning technology. This method solves the challenging phase matching and time-consuming design problems in the process of creating a dielectric lens. It is possible to quickly manufacture arbitrary dielectric lenses and dielectric metasurfaces by using this technology.

## Figures and Tables

**Figure 1 materials-16-02254-f001:**
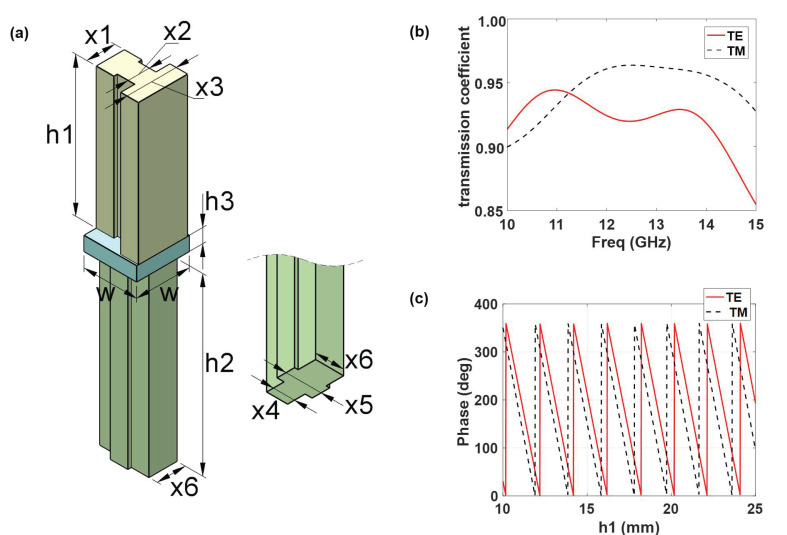
Schematic diagram of unit structure. (**a**) Schematic diagram of unit structure and parameters. (**b**) Anisotropic performance of unit structure. (**c**) Cell structure transmission phase distribution.

**Figure 2 materials-16-02254-f002:**
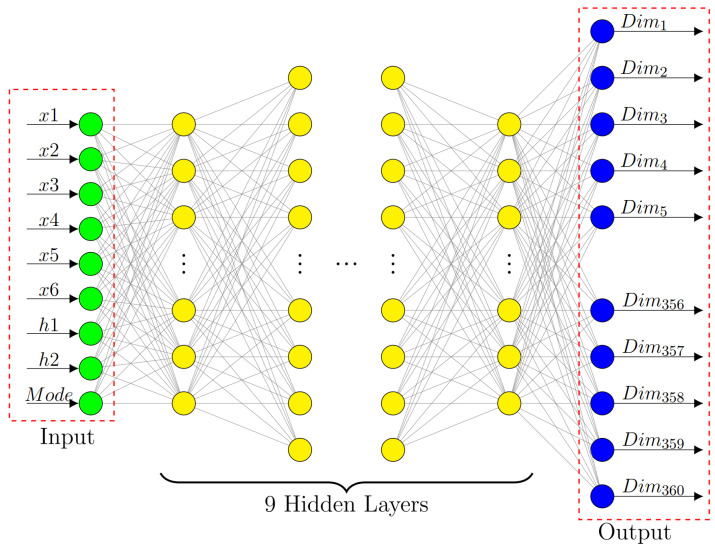
DNN network structure and input/output parameters.

**Figure 3 materials-16-02254-f003:**
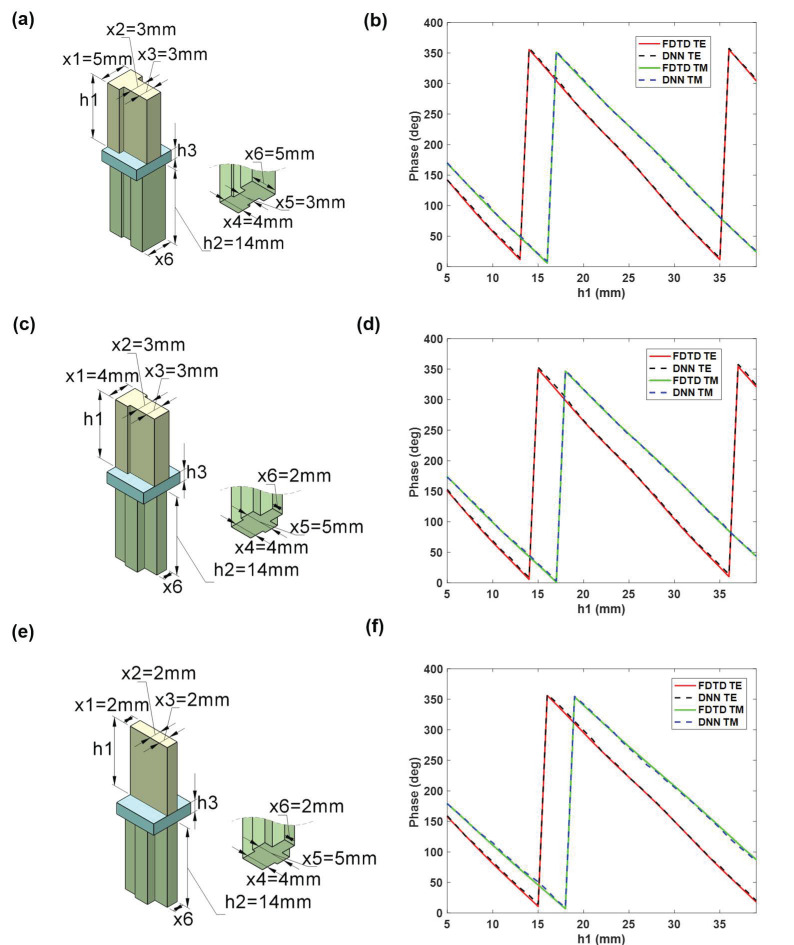
Neural network prediction results comparison chart. (**a**,**c**,**e**) Structure of unit cell chosen at random. (**b**,**d**,**f**) Comparison results of algorithm-based calculation and prediction results.

**Figure 4 materials-16-02254-f004:**
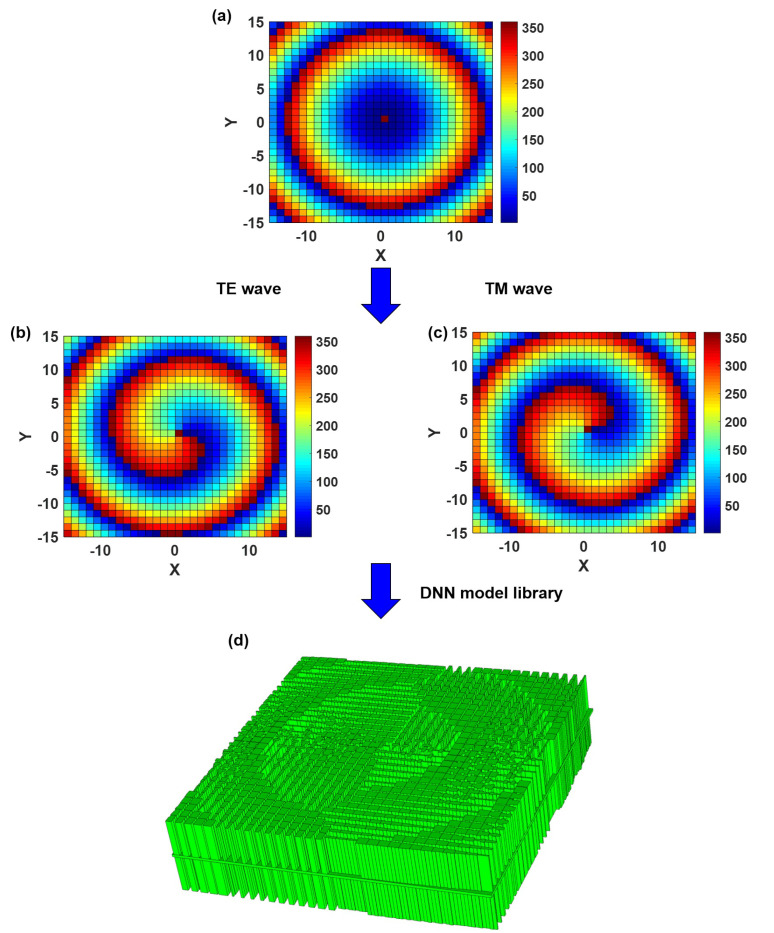
The calculation compensation phase distribution and process on the dielectric lens. (**a**) Phase compensated for feed spherical. (**b**) Compensation phase distribution required to generate l=1 mode vortex wave after TE wave irradiation. (**c**) Compensation phase distribution required to generate l=−1 mode vortex wave after TE wave irradiation. (**d**) Dielectric lens structures generated by machine learning.

**Figure 5 materials-16-02254-f005:**
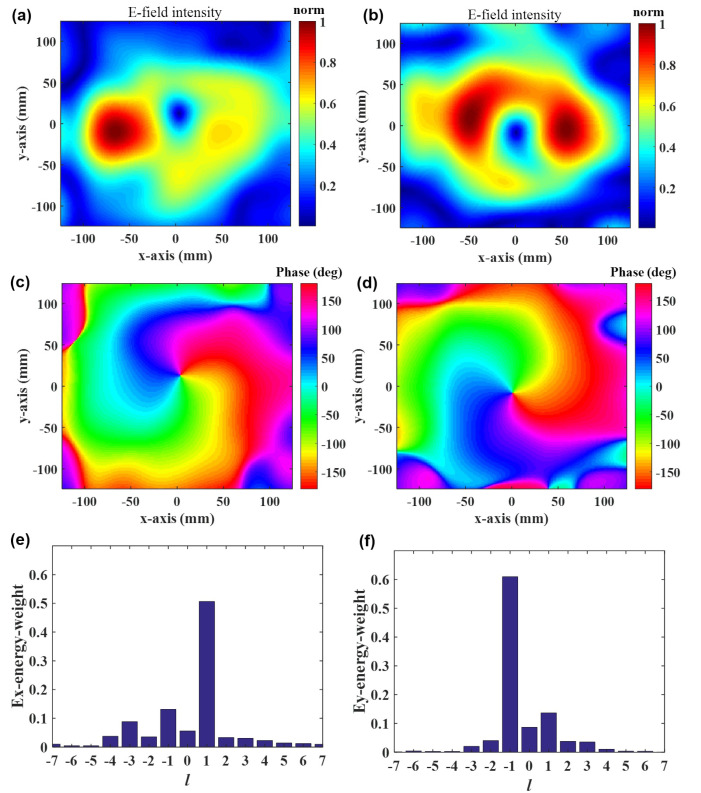
Near-field observation and corresponding spectral analyses at the plane of z=414 mm with a dimension of 248 mm × 248 mm. (**a**) Normalized electrical intensity distribution in x direction under y polarization (TE wave). (**b**) Normalized electrical intensity distribution in x direction under x polarization (TM wave). (**c**) Phase of y-polarization component Ex. (**d**) Phase of x-polarization component Ey. (**e**) Histograms of the simulated OAM spectrum weight for OAM mode number l=+1. (**f**). Histograms of the simulated OAM spectrum weight for OAM mode number l=−1.

**Figure 6 materials-16-02254-f006:**
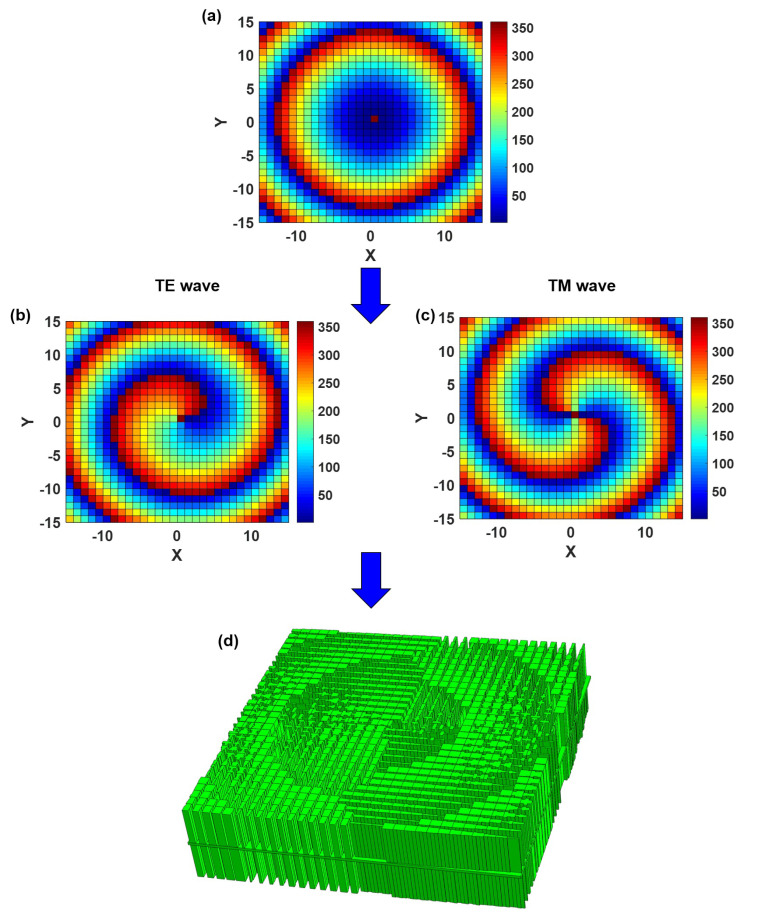
The calculation compensation phase and process on the dielectric lens. (**a**) Phase compensated for feed spherical. (**b**) Compensation phase distribution required to generate l=−1 mode vortex wave after TE wave irradiation. (**c**) Compensation phase distribution required to generate l=+2 mode vortex wave after TE wave irradiation. (**d**) Dielectric lens structures generated by machine learning.

**Figure 7 materials-16-02254-f007:**
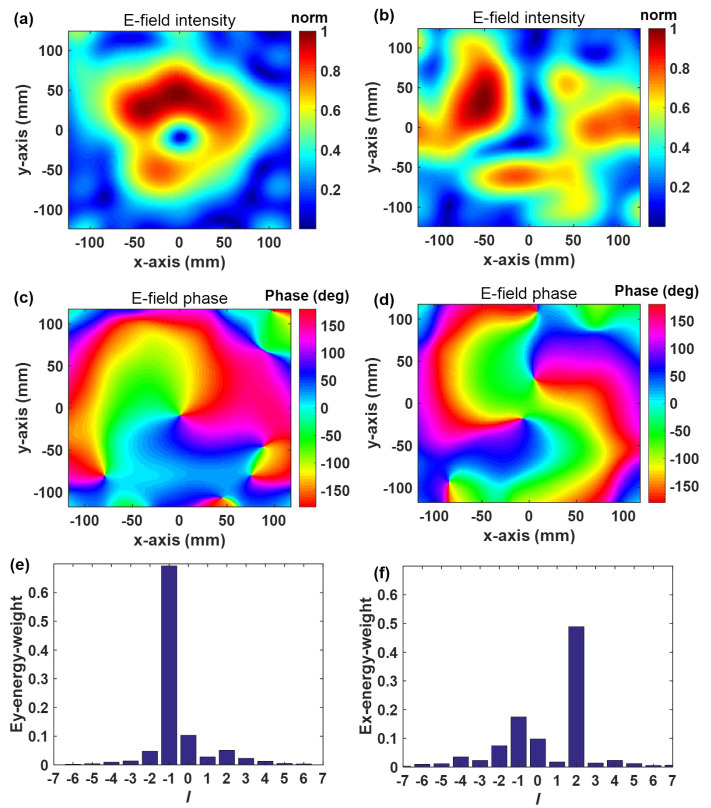
Near-field observation and corresponding spectral analyses at the plane of z=414 mm with a dimension of 248 mm × 248 mm. (**a**) Normalized electrical intensity distribution in x direction under y polarization (TE wave). (**b**) Normalized electrical intensity distribution in x direction under x polarization (TM wave). (**c**) Phase of y-polarization component Ex. (**d**) Phase of x-polarization component Ey. (**e**) Histograms of the simulated OAM spectrum weight for OAM mode number l=−1. (**f**) Histograms of the simulated OAM spectrum weight for OAM mode number l=+2.

**Table 1 materials-16-02254-t001:** DNN network model papameters.

Layer	Neurons	Activation Function
Input	10	–
Hidden layer 1	110	Tanh
Hidden layer 2	100	LeakyReLU
Hidden layer 3	90	LeakyReLU
Hidden layer 4	80	LeakyReLU
Hidden layer 5	70	LeakyReLU
Hidden layer 6	60	LeakyReLU
Hidden layer 7	50	LeakyReLU
Hidden layer 8	40	LeakyReLU
Hidden layer 9	360	LeakyReLU
output	360	Tanh

## Data Availability

Not applicable.

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
