# Peer review of "A Fast Design Method of Anisotropic Dielectric Lens for Vortex Electromagnetic Wave Based on Deep Learning"

_materials, 2023, doi:10.3390/ma16062254_

Round 1

Reviewer 1 Report

The authors propose a deep learning neural network-based quick design method for vortex wave dielectric lenses in this paper. They used DNN networks to learn and optimise the 9-variable unit structure. They used the Grey Wolf optimization method to achieve rapid dielectric lens unit positioning. There are also two vortex wave multiplexed dielectric lenses included.

Overall, I think the manuscript is well-written and well-presented, with some originality.

However, I'd like the authors to explain why they chose the grey wolf optimization algorithm over, say, the Alpha wolf, which is the best fittest solution among all possible solutions.

If the authors explain the previous question, their manuscript should be publishable.

Author Response

Response:

We sincerely thank the reviewer’s question. Gray Wolf Optimization is a classic and widely used optimization method, renowned for its straightforward concept, low number of adjustable parameters, ease of implementation, and simplicity of operation. When choosing an optimization algorithm, we chose the most basic algorithm, which may not be the best fittest solution. Further, in developing and optimizing the algorithm, we will choose a superior solution on this premise.

Reviewer 2 Report

The manuscript entitled "A fast design of anisotropic dielectric lens.." by Bingyang Liang et al. is a technically sound numerical neural network study whose goal is to help engineers to design lenses for producing vortex electromagnetic waves. 

The manuscript reports interesting results but I have some issues that, in my opinion, the authors should clarify.

In particular, the authors use the verbs "create" or "build" (see, for instance, the beginning of section 2). It is not clear if the authors physically built the dielectric structure they are investigating or if they setup a numerical model. In the former case, I could ask how they measured the quantities shown in part b) and c) of, say, Figure 1. If they physically built the structure, they should have reported on the experimental techniques to produce the above mentioned plots. Personally, I assume that they mathematically "built" the structure, not physically. But I might be in error and I would like that the authors better clarify this point.

This is my major remarks. There some minor points. 

1- Sometimes the authors use the word "gray" wolf, sometime "grey" wolf. Please, check the spelling.

2- The quantity MSE in the caption of figure 3 is not defined

3- on line 94 of the manuscript the authors write "...adiusted to Tanh..." and the same capitalize word "Tanh" appears in Table 1. As somewhere else the authors use the word "tan" (not capitalized) for the trigonometrical tangent function, it is not clear if Tanh means the hyperbolic tangent function or if it is a family. 

My opinion that the paper should be published without further reviewing if the authors fix the above mentioned points.

Author Response

We thank both reviewers for their careful reviews and constructive comments. Below, we address the comments point by point.

Reviewer

The manuscript entitled "A fast design of anisotropic dielectric lens.." by Bingyang Liang et al. is a technically sound numerical neural network study whose goal is to help engineers to design lenses for producing vortex electromagnetic waves. 

The manuscript reports interesting results but I have some issues that, in my opinion, the authors should clarify.

In particular, the authors use the verbs "create" or "build" (see, for instance, the beginning of section 2). It is not clear if the authors physically built the dielectric structure they are investigating or if they setup a numerical model. In the former case, I could ask how they measured the quantities shown in part b) and c) of, say, Figure 1. If they physically built the structure, they should have reported on the experimental techniques to produce the above mentioned plots. Personally, I assume that they mathematically "built" the structure, not physically. But I might be in error and I would like that the authors better clarify this point.

This is my major remarks. There some minor points. 

Response:

We sincerely thank the reviewers for pointing this out.

We have designed a unit structure based on the electromagnetic parameters of materials. The overall performance of this unit structure shows the specific electromagnetic performance. So, we built a structure mathematically.

We replaced all the words "created" with "designed", on the Page 2, Line 72, Line 83; Page 3,

line 101; Page 10, Line 226 ,227; Page 11, Line 238.

1- Sometimes the authors use the word "gray" wolf, sometime "grey" wolf. Please, check the spelling.

Response:

We sincerely thank the reviewers for pointing this out. We have updated all the word use "gray" wolf in the revised manuscript.

2- The quantity MSE in the caption of figure 3 is not defined

Response:

In the revised manuscript, we have added a detailed definition of MSE, on the Page 5, line 127—Line 128.

3- on line 94 of the manuscript the authors write "...adiusted to Tanh..." and the same capitalize word "Tanh" appears in Table 1. As somewhere else the authors use the word "tan" (not capitalized) for the trigonometrical tangent function, it is not clear if Tanh means the hyperbolic tangent function or if it is a family. 

My opinion that the paper should be published without further reviewing if the authors fix the above mentioned points.

Response:

The hyperbolic tangent function (Tanh) is one type of activation functions used in deep learning. Of course, some of its variants are often used in the deep learning. In our manuscript, the most basic definitions are used, as follows:

Other functions in the text "tan" (not capitalized) refer to tangent functions.

In the revised version, we have added a detailed description of functions, on the Page 4, line 111—Line 112.

Reviewer 3 Report

This paper implements the fast design of anisotropic dielectric antennas using deep neural networks (DNN). The work done lacks novelty and does not meet the standards of the materials journal to be published in the journal.

1- What is the originality of the article? It should be clearly stated in the introduction.

2- It is necessary to review the references provided and include some more in relation to the proposed work. Highlighting the novelty of the proposed work.

3- The results obtained should be compared with the bibliography introduced, highlighting the novelty and improvements of the proposed work.

4- Which fdtd software was used for the simulation?

Author Response

We thank both reviewers for their careful reviews and constructive comments. Below, we address the comments point by point.

Reviewer

This paper implements the fast design of anisotropic dielectric antennas using deep neural networks (DNN). The work done lacks novelty and does not meet the standards of the materials journal to be published in the journal.

  • What is the originality of the article? It should be clearly stated in the introduction.

Response:

We sincerely thank the reviewer’s suggestions.  

Typically, only one mode of the vortex wave can be realized once the lens structure has been established. An anisotropic lens construction can be utilized to realize more vortex wave modes to enhance the vortex wave's performance. However, during the anisotropic structure design process, more dielectric unit structures need to be estimated. The typical full-wave electromagnetic computation method is time-consuming.

We built a deep learning neural network that can realize the rapid design of multiple anisotropic dielectric lenses to realize additional modes of vortex wave in the same dielectric structure and improve its communication capacity. 

In the revised version, we have added the relevant contents, on the Page 2, line 46-51, Line 75-76.

2- It is necessary to review the references provided and include some more in relation to the proposed work. Highlighting the novelty of the proposed work.

Response:

 Qian Zhang et al. proposed using machine learning to design 1-bit anisotropic digital Coding Metasurfaces. It is very convenient to realize the left and right circularly polarized medium metasurfaces[a1]. Nathan Bryn et al. proposed a deep learning technique for the forward and inverse plasmonic metasurface structure designs [a2]. The machine learning algorithm used above to build metasurfaces has produced successful outcomes, the needed cell can be designed in milliseconds. However, for anisotropic vortex electromagnetic dielectric lenses that need more dielectric unit structures, there hasn't been any work to bring the machine learning-based design process into practice.

In the revised version, we have added the relevant contents, on the Page 2, line 61-65; 70-73.

  1. The results obtained should be compared with the bibliography introduced, highlighting the novelty and improvements of the proposed work.

Response:

The accuracy of the neural network we presented can be improved by 2-7% compared to the existing DNN network, and there are more free parameters [a1-a3]. In the revised version, we have added the relevant contents, on the Page 5, line 131-133.

When compared to the conventional approach, the performance of the machine learning-designed dielectric lens has surpassed that of the conventional approach in several areas (purity, anisotropic), and it is designed considerably more quickly [a4-a5].

In the revised version, we have added the relevant contents, on the Page 10, line 225-228.

4- Which fdtd software was used for the simulation?

Response:

The FDTD software in this study is mainly CST STUDIO SUITE, and some self-developed software packages.

[a1] Zhang Q, Liu C, Wan X, Zhang L, Liu S, Yang Y, Cui TJ. Machine‐learning designs of anisotropic digital coding metasurfaces. Advanced theory and simulations. 2019 Feb;2(2):1800132
[a2] Roberts, Nathan Bryn, and Mehdi Keshavarz Hedayati. "A deep learning approach to the forward prediction and inverse design of plasmonic metasurface structural color." Applied Physics Letters 119.6 (2021): 061101.
[a3] Jiang, Li, et al. "Neural network enabled metasurface design for phase manipulation." Optics Express 29.2 (2021): 2521-2528.
[a4]  Qin, Fan, et al. "A transmission metasurface for generating OAM beams." IEEE Antennas and Wireless Propagation Letters 17.10 (2018): 1793-1796.
[a5] Lin, Zhansong, Zhongling Ba, and Xiong Wang. "Broadband high-efficiency electromagnetic orbital angular momentum beam generation based on a dielectric metasurface." IEEE Photonics Journal 12.3 (2020): 1-11.
